# Citance-Contextualized Summarization of Scientific Papers

**Shahbaz Syed** [†*]    **Ahmad Dawar Hakimi** [†*]    **Khalid Al-Khatib** [‡]    **Martin Potthast** [†§]

[†]Leipzig University        [‡]University of Groningen        [§]ScaDS.AI

shahbaz.syed@uni-leipzig.de

## Abstract

Current approaches to automatic summarization of scientific papers generate informative summaries in the form of abstracts. However, abstracts are not intended to show the relationship between a paper and the references cited in it. We propose a new contextualized summarization approach that can generate an informative summary conditioned on a given sentence containing the citation of a reference (a so-called "citance"). This summary outlines the content of the cited paper relevant to the citation location. Thus, our approach extracts and models the citances of a paper, retrieves relevant passages from cited papers, and generates abstractive summaries tailored to each citance. We evaluate our approach using WEBIS-CONTEXT-SCISUMM-2023, a new dataset containing 540K computer science papers and 4.6M citances therein.[1] [2]

## 1 Introduction

The original task of automatic summarization has been the abstracting of scientific papers, one of the first tasks studied in computer science (Luhn, 1958; Baxendale, 1958). Automatically generated abstracts were used to create "index volumes" for specific scientific areas to help researchers access the growing number of publications. Nowadays, paper authors usually write abstracts themselves. However, author-generated abstracts often provide incomplete or biased coverage of scientific papers (Elkiss et al., 2008). As a result, the purpose of automatic paper summarization has evolved to generate more informative summaries, often using abstractive summarization approaches (Cohan et al., 2018; Cachola et al., 2020; Mao et al., 2022).

A practical application of generating summaries is to augment reading papers. For example, CITE-READ by Rachatasumrit et al. (2022) is part of Allen AI's Semantic Reader (Lo et al., 2023) and shows on demand the abstracts as summaries for the cited papers in a paper being read. While these

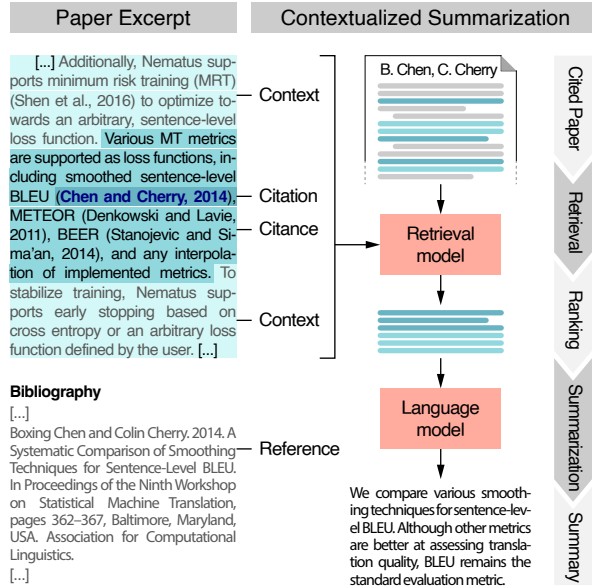

Figure 1: Contextualized summarization at a glance: Given a citation, our approach uses the citance and its contexts as a queries to retrieve relevant content from the cited paper. A contextualized summary is then generated using a large language model.

abstracts provide a concise and general overview of the cited papers, they do not meet the information need of readers trying to understand the relevance of a paper in the context of its citation. The generated abstracts are not adapted to the citation context, leaving the reader to consult the cited work directly.

In this paper, we investigate the suitability of contextualized summaries that are specifically tailored to individual citation contexts compared to generic abstracts. We propose a new approach to generating contextualized summaries by operationalizing citation contexts. Figure 1 illustrates our approach, which consists of three steps (Section 3): (1) Extraction and modeling of the sentence containing a citation (the citance), and its contexts from the citing document, (2) retrieval of relevant content from the citing paper using queries based on these citance contexts, and (3) generation of abstractive, citance-contextualized summaries of the citing paper. To address this novel task, we create WEBIS-CONTEXT-SCISUMM-2023 (Sec-

---

* Equal contribution.
[1]Code: https://github.com/webis-de/EMNLP-23
[2]Data: https://zenodo.org/doi/10.5281/zenodo.10031025

tion 4), a large-scale, high-quality corpus consisting of 540K computer science papers and 4.6M citances. In an extensive comparative evaluation using our corpus, we explore different variants of our approach in comparison to the abstracts of the cited papers (Section 5). We find that while abstracts have a slight advantage in terms of coverage and focus, when a citance fails to align with the central theme of the cited paper, contextualized summaries prove to be a more favorable alternative to abstracts. Examples of contextualized summaries from our approach along with the abstracts are shown in Appendix D.

## 2 Related Work

In this section, we review the literature on generic and citation-based summarization of scientific documents, including different types of generated summaries, and approaches that summarize the target paper-based on its citance (contexts), respectively.

### 2.1 Generic Summarization

Generic summarization approaches for scientific papers are based on various ground-truth summaries, including abstracts (Luhn, 1958; Cohan et al., 2018), author-written highlights (Collins et al., 2017), author-written promotional blurbs (Chen et al., 2020), and condensed versions of summaries from peer reviews (Cachola et al., 2020).

**Abstract-based summarization** Collins et al. (2017) proposed a supervised model for extractive summarization trained on 10,148 computer science papers. The model uses an LSTM-based neural encoder with lexical features to classify summary-worthy sentences, where author-written highlights and abstracts serve as references. Cohan et al. (2018) presented a discourse-aware attention model for the abstractive summarization of scientific papers from the arXiv and PubMed collections. A hierarchical encoder integrates section information to generate coherent summaries. Gupta et al. (2021) investigated pre-training and tuning BERT-based models for extractive summarization.

**TL;DR summarization** Ultra-short indicative TL;DR summaries are concise, typically one or two sentences short, and aim to highlight the key findings of a paper. Cachola et al. (2020) developed the SCITLDR corpus, consisting of 3.2K papers with accompanying manually written TL;DR summaries (15-25 words) sourced from peer reviews

and from the authors of papers. Control codes and multitask learning were used to generate summaries; the model also used the paper titles as an additional training signal.

**Comprehensive summarization** LongSumm is a task, which aims to generate comprehensive summaries of about 600 words, providing sufficient information in lieu of reading the target paper and overcoming the limitations of abstracts and TL;DR summaries Chandrasekaran et al. (2020). The LongSumm corpus comprises 2236 papers with abstractive and extractive summaries. Sotudeh et al. (2021) created two corpora from arXiv and PubMed consisting of 11,149 and 88,035 paper–summary pairs, respectively. To guide the generation of the long summaries, Sotudeh and Goharian (2022) expanded abstracts with sentences from introduction, overview, and motivation sections.

### 2.2 Citation-based Summarization

In citation-based summarization, citances from the source paper are used as queries to extract relevant content from the target paper, and to generate a summary. Qazvinian and Radev (2008) analyzed the citation network of target papers and collected citances from different sources. These citances were clustered, and the central sentences identified as extractive summaries. Mei and Zhai (2008) focused on impact sentence-based summaries, reflecting the authority and proximity of the citations in a paper collection. The impact of the target paper on related work is determined using citations from the source papers. To improve readability and coherence, Abu-Jbara and Radev (2011) introduced a preprocessing step to filter out non-relevant text segments. Then, an extraction phase is conducted to select important sentences from sections such as background, problem statement, method, results, and limitations. In a post-processing step, the overall readability of generated summaries is improved, replacing pronouns and resolving co-references.

Closely related to our work, Cohan and Goharian (2015) used citance contexts, defined as the text passages of the target paper that reflect the citation of the source paper. To summarize the target paper, they first collected multiple citance contexts, constructing a graph based on their intra-connectivity based on the cosine similarity of tf-idf vectors. The sentences in this graph were ranked by their importance (number of connections). The retrieved sentences are combined with discourse information

from the target paper to generate an informative summary. Cohan and Goharian (2017) have further improved this model using word embeddings and domain knowledge to enhance the citation contexts.

Our work also focuses on contextualizing citations using citance contexts, but differs significantly: We identify different types of citance contexts and use them to generate multiple contextually relevant summaries for a given citance. Instead of relying solely on the literal citance as a query, which represents only one type of citance context, we employ multiple contexts for deriving queries.

Our corpus is the largest one with citance context-specific summaries of scientific papers, comprising about 540,000 papers and 4.6 million citances. In comparison, the CITESUM corpus of Mao et al. (2022) comprises only 93,000 papers, where a citance from the related work section of the source paper serves as an ultra-short summary of the target paper. Our corpus includes citances from all sections of the source paper and contains multiple types of citance contexts, as well as multiple summaries for each context. As a result, our corpus provides a comprehensive and diverse resource for studying the summarization of scientific papers. g

## 3 Contextualized Summarization

Our approach to contextualized summarization involves using multiple citance contexts in a source paper. In addition to the citance itself (a single sentence containing the citation), we consider several types of surrounding contexts. As illustrated in Figure 1, our approach involves three main steps: (1) extraction of citances, (2) retrieval of relevant content from the cited paper, and (3) generation of abstractive summaries that are contextualized based on a citance.

### 3.1 Extraction of Citance-contexts

First, all citances that literally refer to other papers are extracted from a given paper. We then consider two additional contexts for a citance. The first includes the sentences that immediately precede and follow the citance. The second contains the two semantically most similar sentences of the citance within the same paragraph. This yields three citance contexts: (1) the *citance* itself, (2) the citance and its *neighbors*, and (3) the citance and semantically *similar* sentences. By considering these contexts, we aim to improve the retrieval of relevant content from the cited paper.

### 3.2 Citance-contextualized Retrieval

We use the above three citance contexts as queries for retrieval. In addition, we explore the use of extracted keywords from each citance context to improve the queries (Carpineto and Romano, 2012). For retrieval, we use both shallow and dense retrieval models (Section 4.2). We retrieve relevant content at two levels of granularity: sentences and paragraphs. Specifically, we extract the top-5 relevant sentences and the top-2 relevant paragraphs from the cited paper. This enables the evaluation of which granularity is more suitable for the contextualized summarization task.

The top-5 most relevant sentences provide a broader *coverage* of the cited paper that includes information relevant to the citance. Conversely, the top-2 most relevant paragraphs provide a higher degree of *focus*, where the summary sentences are interconnected. Therefore, we experiment with both granularities to investigate their effectiveness in our approach. Following the retrieval process, we perform a qualitative evaluation of the retrieved content. This evaluation helps us select the optimal combination of query and retrieval models for the subsequent summarization step (Section 5.1).

### 3.3 Citance-contextualized Summarization

After retrieving the relevant content from the cited paper, we use it as input to the summarization model. This ensures that the generated summaries are thus contextualized to the citance and focus exclusively on the parts of the cited paper that are relevant to it. In our approach, we explore the effectiveness of large language models (LLMs) due to their strong multi-task capabilities (Bommasani et al., 2021). We use prompt-based, instruction-tuned models that can understand and execute natural language instructions from the user to accomplish a specific task. The flexibility and adaptability to different domains distinguishes our approach from domain-specific supervised methods.

Since we have two granularities of input to the summary model (top-5 sentences and top-2 paragraphs), we design two prompts tailored to both. For the top-5 sentences, we use a paraphrasing prompt that aims to transform the sentences into a coherent summary. For the top-2 paragraphs, we use an abstractive summarization prompt to generate a coherent summary from them. For more details on the prompts, see Section 4.4.

| Corpus | Size | Avg. Length | Style | Type | Multiple |
|---|---|---|---|---|---|
| LaySumm (Chandrasekaran et al., 2020) | 572 | 84 tokens | Layman-Summary | Abs. | ✗ |
| SciSummNet (Yasunaga et al., 2019) | 1K | 151 words | Abstract | Ext. | ✗ |
| TalkSumm (Lev et al., 2019) | 1.7K | 965 words | Informative | Ext. | ✗ |
| LongSumm (Chandrasekaran et al., 2020) | 2.2K | 779 words | Blog Post | Ext./Abs. | ✗ |
| SciTLDR (Cachola et al., 2020) | 3.2K | 21 words | TL;DR | Abs. | ✓ |
| FacetSum (Meng et al., 2021) | 60K | 290 words | Facet-oriented, Informative | Ext./Abs. | ✓ |
| CiteSum (Mao et al., 2022) | 93K | 23 words | TL;DR | Abs. | ✗ |
| CORD-SUM (Qi et al., 2022) | 123K | 223 words | Abstract | Ext. | ✗ |
| PubMed (Cohan et al., 2018) | 133K | 203 words | Abstract | Abs. | ✗ |
| ArXiv (Cohan et al., 2018) | 215K | 220 words | Abstract | Abs. | ✗ |
| RSCSum (Chen et al., 2020) | 308K | 29 words | Table of Contents | Abs. | ✗ |
| ArXiv-Long (Sotudeh and Goharian, 2022) | 11.1K | 574 tokens | Extended Abstract | Ext. | ✗ |
| PubMed-Long (Sotudeh and Goharian, 2022) | 88K | 403 tokens | Extended Abstract | Ext. | ✗ |
| WEBIS-CONTEXT-SCISUMM-2023 | 540K | 117 tokens | Citance-contextualized, Informative | Abs. | ✓ |

Table 1: Comparison of our WEBIS-CONTEXT-SCISUMM-2023 corpus with existing corpora for summarizing scientific papers. The columns show the size of each corpus, the characteristics of their summaries, such as the average length in tokens or words, the target style, the type of summary (extractive or abstractive), and whether there are multiple summaries per document in the corpus. Our corpus provides a unique combination of multiple, abstractive, citance-contextualized, and informative summaries of a cited paper. The average length of the summaries exceeds the 100 references produced for qualitative evaluation (Section 5).

## 4 WEBIS-CONTEXT-SCISUMM-2023: A Large-Scale Corpus for Contextualized Summarization of Scientific Papers

Previous datasets for summarizing scientific papers do not consider different types of citance contexts, nor do they evaluate multiple retrieval models for extracting relevant content (Section 2). Therefore, these datasets are not suitable for studying citance-contextualized summarization. To address this gap, we introduce WEBIS-CONTEXT-SCISUMM-2023, a new and large dataset created using our approach as described in Section 3.

### 4.1 Data Source and Preprocessing

We used the publicly available Semantic Scholar Open Research Corpus (S2ORC) (Lo et al., 2020).[3] The corpus consists of 136 million scientific documents, of which 12 million are available in full text. We focused on a subset of 870,000 documents from the field of computer science (Section 5.2). After documents without citations were removed, about 540,000 documents remained. We then extracted citances by identifying the sentences in each document containing citations. This resulted in a total of 4.6 million citances. Unlike Mao et al. (2022), who only considered citances from the *Related Work* section, we considered citances from all paper sections, resulting in a more diverse set.

### 4.2 Citance-contexts and Retrieval Models

As described in Section 3.1, we employ three types of citance contexts as queries to retrieve relevant content from a cited paper. The *citance* and the *neighbors* contexts are extracted directly. For the *similar* context, the contextual embeddings from SciBERT (Beltagy et al., 2019) were used to identify the two semantically most similar sentences of a citance using the cosine similarity.[4] We also extracted keywords as queries from the contexts using KeyBERT (Grootendorst, 2020).

As retrieval models, we used BM25 (Robertson et al., 1994)[5] and the cosine similarity of SciBERT embeddings between the query (citance context) and the document (sentences or paragraphs of the cited paper) to contrast both shallow and dense retrieval paradigms. The combination of the three types of queries (including keyword variants) and the two retrieval models resulted in a total of 12 retrieval setups, as shown in Table 3, along with their mean NDCG@5 scores from our internal evaluation (Section 5.1). We indexed 151 million sentences and 40 million paragraphs to retrieve the top-5 sentences and the top-2 paragraphs, respectively, for each query. For the keyword queries, we used weighted aggregation to fuse the individual

---

[3]We used the S2ORC dataset released on 2020-07-05.

[4]`scibert-scivocab-uncased` from https://huggingface.co/allenai/scibert_scivocab_uncased

[5]We used the *Rank-BM25* toolkit (Brown, 2020) with default parameters for BM25 ($k$=1, $b$=0.75).

| Model Variant | Description |
|---|---|
| Alpaca 7B | LLaMA-7B model finetuned on 52K examples of self-instructed instruction-following responses (Wang et al., 2022). |
| Vicuna 13B | LLaMA models finetuned on user-shared conversations collected from ShareGPT.[6] |
| LLaMA-CoT | LLaMA-30B model finetuned on chain-of-thought and logical deduction examples (Qingyi Si, 2023). |
| Falcon Instruct | Trained on the RefinedWeb corpus (Penedo et al., 2023), derived through extensive filtering and deduplication of publicly available web data. |
| GPT4 | The latest version of the GPT model from OpenAI.[7] We use it to bootstrap ground-truth summaries (Section 5.2). |

Table 2: List of LLMs used for zero-shot abstractive summarization. See Appendix C for more details.

rankings: Each ranking's weight corresponded to a query's cosine similarity to its citance; the resulting rankings were combined by a weighted sum.

### 4.3 Corpus Statistics

The compiled corpus consists of 537,155 computer science papers containing a total of 4,619,552 citances. On average, each paper contains 8.6 citances. The mean length of a citance is 31 tokens, and the median is 27 tokens. In addition, the corpus contains 346,450 papers that have multiple citances to the same target paper, facilitating the study of contextualized summarization approaches. Table 1 compares our corpus to other datasets.

### 4.4 Abstractive Summarization

Using the retrieved content from the cited paper, we used the prompt-based instruction-tuned LLMs listed in Table 2 for abstractive summarization of each citance. For the two granularities of retrieved content (top-5 sentences and top-2 paragraphs), we generated separate summaries using the models in a zero-shot setting. For the top-5 sentences, we paraphrased them into coherent text since they already served as extractive summaries. For the top-2 paragraphs, we performed abstractive summarization. Throughout the tasks, we experimented with different instructions and prompt formulations.

#### 4.4.1 Prompt Formulation

To generate text conditioned on specific instructions, the above models require a tailored prompt. We conducted experiments with different instructions and prompt formulations for paraphrasing

**Paraphrasing Prompt**

**### Instruction:**
A chat between a curious human and an artificial intelligence assistant. The assistant knows how to paraphrase scientific text and the user will provide the scientific text for the assistant to paraphrase.
**### Input:**
Generate a coherent paraphrased text for the following scientific text: {*input*}.
**### Output:**

**Summarization Prompt**

**### Instruction:**
A chat between a curious human and an artificial intelligence assistant. The assistant knows how to summarize scientific text and the user will provide the scientific text for the assistant to summarize.
**### Input:**
Generate a coherent summary for the following scientific text in not more than 5 sentences: {*input*}.
**### Output:**

Figure 2: Best prompts with instructions for paraphrasing (the top-5 retrieved sentences) and summarizing (the top-2 retrieved paragraphs). We ensured that the summaries for both granularities were similar in length by instructing the model not to generate more than five sentences for the top-2 paragraphs.

and summarizing. The generated summaries for 10 examples from all models were evaluated manually. Based on this evaluation, the best combination of instructions and prompt formulations for each model was selected. Figure 2 shows the selected combination, and Appendix A gives more details.

## 5 Evaluation

We evaluated the retrieval step of our approach in an internal evaluation, and the summarization step in an external one.

### 5.1 Content Retrieval Evaluation

The 12 retrieval setups shown in Table 3 were evaluated using manual relevance assessments of the content retrieved from cited papers to the corresponding citance context (query) of citing papers. Ten queries were used to retrieve the top-5 sentences of a cited paper for each of the 12 setups, resulting in a total of 600 sentences. Sentence relevance was assessed on a graded scale: *relevant*, *somewhat relevant*, and *non-relevant*. Table 3 shows the results in terms of NDCG@5 (Järvelin and Kekäläinen, 2002). Based on them, we selected the *similar* context as query for BM25 and

| BM25 (Shallow) | | SciBERT (Dense) | |
|---|---|---|---|
| Query | Mean nDCG@5 | Query | Mean nDCG@5 |
| citance | 0.943 | citance | **0.943** |
| similar | **0.958** | similar | 0.918 |
| neighbors | 0.898 | neighbors | 0.801 |
| citance-keywords | 0.914 | citance-keywords | 0.617 |
| similar-keywords | 0.944 | similar-keywords | 0.650 |
| neighbors-keywords | 0.928 | neighbors-keywords | 0.706 |

Table 3: Evaluation of 12 retrieval setups as combinations of a shallow and a dense retrieval model with citance contexts as queries to extract relevant content from cited papers. We report mean nDCG@5 for 600 relevance judgments. The best combination (in bold) has been selected for the summarization step.

the *citance* context as query for SciBERT as best setups for shallow and dense retrieval to evaluate the subsequent summarization step. The former uses the top-2 semantically most *similar* sentences to a citance (along with the citance itself) as a query, while the latter uses the *citance* only.

## 5.2 Summarization Evaluation

The contextualized summaries of the models listed in Section 4.4 were evaluated using both quantitative and qualitative methods. For quantitative evaluation, we used the ROUGE (Lin, 2004) and BERTScore (Zhang et al., 2020) metrics. For qualitative evaluation, we manually scored the top two models in terms of coverage and focus.

**Evaluation Data** Fifteen articles were selected from the ACL anthology, published between 2016 and 2020. We extracted 363 citances from these articles and randomly selected 25 of them. Using the full texts of the cited papers and the top two retrieval models from Table 3, the top-5 sentences and the top-2 paragraphs were retrieved, resulting in a total of 100 texts. To create the ground-truth reference summaries, we used GPT4 (Bubeck et al., 2023) in a zero-shot setting to paraphrase/summarize these texts using the prompts shown in Figure 2. Each summary was then manually reviewed to ensure accuracy and to rule out hallucinations or factual errors. Our set of references consists of 100 summaries (= 25 citances × 2 retrieval models × 2 summary types).

**Automatic Evaluation** The reference summaries were used to automatically evaluate the generated contextualized summaries. Table 4 shows the results. According to ROUGE, Vicuna performs

| Model | BERTScore | ROUGE | | |
|---|---|---|---|---|
| | | R-1 | R-2 | R-L |
| **top-2 paragraphs** | | | | |
| *similar-BM25* | | | | |
| Alpaca | 0.343 | 47.3 | 25.5 | 44.9 |
| Falcon | 0.401 | 48.2 | 27.1 | 45.0 |
| LLaMA-CoT | 0.448 | 53.0 | 31.9 | 50.5 |
| Vicuna | 0.465 | **58.7** | **35.4** | **55.8** |
| *citance-SciBERT* | | | | |
| Alpaca | 0.390 | 54.3 | 32.2 | 52.0 |
| Falcon | 0.413 | 52.1 | 29.6 | 48.9 |
| LLaMA-CoT | **0.497** | 54.7 | 32.9 | 52.5 |
| Vicuna | 0.431 | 56.7 | 34.2 | 53.9 |
| **top-5 sentences** | | | | |
| *similar-BM25* | | | | |
| Alpaca | 0.616 | 56.2 | 35.4 | 54.8 |
| Falcon | 0.649 | 57.5 | 35.6 | 55.2 |
| LLaMA-CoT | 0.707 | 61.2 | 38.6 | 60.0 |
| Vicuna | 0.551 | 57.2 | 34.3 | 54.9 |
| *citance-SciBERT* | | | | |
| Alpaca | 0.595 | 56.6 | 34.7 | 55.1 |
| Falcon | 0.656 | 56.8 | 36.2 | 55.3 |
| LLaMA-CoT | **0.748** | **62.9** | **40.6** | **60.9** |
| Vicuna | 0.607 | 58.8 | 36.0 | 56.6 |

Table 4: Automatic evaluation of summaries from all LLMs grouped by two granularities: top-2 relevant paragraphs and top-5 relevant sentences from the cited paper. We report BERTScore (accuracy) and different ROUGE scores compared to reference summaries from GPT4. For manual evaluation, we selected the best model from each setup based on ROUGE overlap with the references: Vicuna (*similar-BM25*) and LLaMA-CoT (*citance-SciBERT*) for the top-2 paragraphs and top-5 paragraphs, respectively.

best in summarizing the top-2 paragraphs, while LLaMA-CoT performs best in paraphrasing the top-5 sentences as summaries. Moreover, it also achieves the highest BERTScore in the top-2 paragraph setting. Therefore, we evaluated it manually for coverage and focus.

**Human Evaluation** Three domain experts were recruited, including two students and one post-doc, to evaluate the usefulness of the summaries. The annotators were asked to rate the summaries on the two criteria *coverage* and *focus*. Ratings were on a 5-point Likert-scale, with 1 indicating the worst and 5 the best rating. Coverage reflects how well the summary captures the essential information from the cited paper that is relevant to a particular citance, whereas *focus* refers to the coherence and cohesion of the sentences in the summary. A total of 125 summaries of 25 cited papers were evaluated. Each sample consisted of the citance (and its context) displayed on the left-hand side

You are a scientist who is currently reading a paper. While reading the paper, you see a citation to another paper that you want to follow. You are also given a summary of the corresponding cited paper. Your task is to assess is to rate this summary on {coverage/focus/relevance}. Please make sure you read and understand these instructions carefully. Please keep this document open while reviewing, and refer to it as needed.

**Evaluation Criteria**:

*Coverage* (1-5) - the amount of key information covered by the summary that is relevant to this citation. The summary should contain information from the cited paper that fits the context of the given citation text and helps you to better understand why this paper was cited here.

*Focus* (1-5) - the collective quality of all sentences. The summary should be well-structured and well-organized. The summary should build from sentence to sentence to a coherent body of information that is relevant to the given citation text.

*Relevance* (1-5) - the relevance of the summary to the citation text. The summary should contain only information from the cited paper that fits the current reading context. It should be sufficiently informative so that you do not need to actually read the cited paper to understand why it was cited here.

**Evaluation Steps (Corrected Chain of Thought)**:

1. Read the given citation text that references another paper and make note of its content.

2. Read the summary provided of the cited paper.

3. Coverage:

    (a) Check if the summary contains only information that is relevant to the citation text.
    (b) Assign a score for coverage on a scale of 1 to 5, where 1 is the lowest and 5 is the highest based on the Evaluation Criteria.

4. Focus:

    (a) Check if the summary is coherent and contains well-organized information that is relevant to the citation text.
    (b) Assign a score for focus on a scale of 1 to 5, where 1 is the lowest and 5 is the highest based on the Evaluation Criteria.

5. Relevance:

    (a) Compare the content of the summary with the citation text. Assess whether the summary includes relevant information from the cited paper that directly relates to the context and purpose of the citation.
    (b) Determine if the summary provides enough information for you to understand why the cited paper was referenced without needing to read the entire paper.
    (c) Rate the relevance of the summary based on the above assessment using a scale of 1 to 5, where 1 indicates low relevance and 5 indicates high relevance.

**Example**:

*Source Text*: {{Citance}}
*Summary*: {{Summary}}

**Evaluation Form (scores ONLY)**:

- Coverage:
- Focus:
- Relevance:

Figure 3: G-Eval (Liu et al., 2023) instructions for automatically evaluating the summary coverage, focus, and relevance using GPT4. Instructions are merged into a single prompt only for illustration purposes and are used separately in the automatic evaluation.

| Summary | Human Eval. | | G-Eval | | |
|---|---|---|---|---|---|
| | Cov. | Focus | Cov. | Focus | Rel. |
| Abstract | **3.67** | **4.50** | **3.12** | **3.80** | **3.23** |
| *similar-BM25, top-2 paragraphs* | | | | | |
| GPT4 (Reference) | 2.92 | 3.83 | 2.60 | 3.46 | 3.10 |
| Vicuna | 3.01 | 3.56 | 2.96 | 3.20 | 2.80 |
| *citance-SciBERT, top-5 sentences* | | | | | |
| GPT4 (Reference) | 2.45 | 2.99 | 2.35 | 3.14 | 2.56 |
| LLaMA-CoT | 2.33 | 2.33 | 2.40 | 3.12 | 2.63 |

Table 5: Scores for summary quality criteria averaged over 125 summaries according to human evaluation and automatic evaluation with G-Eval. We also automatically evaluated the relevance of a summary to a citance. Summary types are grouped by retrieval setup.

and five summaries on the right-hand side: the abstract of the cited paper, two reference summaries (top-5 sentences and top-2 paragraphs), and the summaries generated by the two best models for the top-5 sentences and top-2 paragraphs. The order of the summaries was randomized to mitigate order effects (Mathur et al., 2017).

**IAA and Results**   Annotators agreement was calculated using Cohen's weighted kappa (Cohen, 1960). We obtained a $\kappa$ of 0.42 and 0.40 for coverage and focus, respectively. While these results indicate some agreement between annotators, they also hint at the inherent subjectivity of this annotation task. Assessing the usefulness of a summary is influenced by several contextual factors, such as the annotators' goals in reviewing a citation, their prior knowledge of the cited paper, and the presentation of the summary (Jones, 2007). In future work, we plan to investigate these factors further.

As shown in Table 5, abstracts as summaries achieved the highest coverage score (3.67), followed closely by the summaries generated by Vicuna (3.01). Abstracts were also rated the best summary for focus (4.50), while the reference summary of GPT4 was only second (3.83). In terms of granularity of retrieved content, the summary based on the top-2 paragraphs outperformed the summary based on the top-5 sentences in terms of both coverage and focus.

However, despite the general preference for abstracts over generated contextualized summaries, annotators provided feedback that our summaries were more effective when a citance was ambiguous and did not relate to the overall idea of a paper. In such cases, they preferred our retrieval-based

summaries over abstracts. Table 6 shows examples of self-contained and ambiguous citances and annotator preferences. To substantiate this result, we plan to extend our evaluation to a larger number of citances in future work.

**LLM-based Evaluation**   To investigate the reliability of evaluating the quality of summaries using LLMs, we used *G-Eval* (Liu et al., 2023), which uses GPT4 to evaluate summary quality based on certain criteria. We evaluated coverage, coherence, and relevance using prompts to assign scores from 1 to 5. *G-Eval* first lets the underlying model to generate a chain of thought to ensure it understands the task. Evaluation instructions along with the (manually) corrected chain of thought for each criterion are illustrated in Figure 3. Table 5 shows the results, which indicate that *G-Eval* reflects the human assessments for the top models, with slight deviations for the lower-ranked ones. Notably, the reference summaries from GPT4 scored similarly to abstracts in terms of relevance.

## 6   Conclusion

We investigated the use of generic paper abstracts compared to tailored contextualized summaries to improve a reader's understanding of the relevance of individual paper citations. For this purpose, we developed a new summarization approach to generate citance-contextualized summaries. With WEBIS-CONTEXT-SCISUMM-2023, we compiled a large-scale corpus to facilitate research in this direction. Experiments with zero-shot summarization using LLMs showed that abstracts are slightly preferred over contextualized summaries in terms of coverage and focus, while summaries generated using our approach are preferred when citances do not refer to the main contribution of a paper.

## 7   Limitations

Our proposed approach for contextualized summarization of scientific papers is based on standard retrieval models and LLMs. It should be noted that what is considered relevant to a citance is subjective and depends on the reader's prior knowledge of the cited paper. As a result, retrieval models may not always retrieve the most relevant context, which may affect the quality of the subsequent abstract. This may explain why abstracts intended to be informative to a broad audience scored better in human evaluations than contextualized summaries.

**Self-contained Citance**

The skill implements Hybrid Code Networks (HCNs) described in (Williams et al., 2017).

**Abstract (✔)**

End-to-end learning of recurrent neural networks (RNNs) is an attractive solution for dialog systems; however, current techniques are data-intensive and require thousands of dialogs to learn simple behaviors. We introduce Hybrid Code Networks (HCNs), which combine an RNN with domain-specific knowledge encoded as software and system action templates. Compared to existing end-to-end approaches, HCNs considerably reduce the amount of training data required, while retaining the key benefit of inferring a latent representation of dialog state. In addition, HCNs can be optimized with supervised learning, reinforcement learning, or a mixture of both. HCNs attain state-of-the-art performance on the bAbI dialog dataset, and outperform two commercially deployed customer-facing dialog systems.

**Vicuna** (Top 2 Relevant Paragraphs Summarized)

This paper presents a new model called Hybrid Code Networks (HCNs) for end-to-end learning. HCNs allow developers to express domain knowledge and achieve better performance with less training data compared to existing techniques. The model is trained using supervised or reinforcement learning and can be applied to the bAbI dialog dataset and real customer support domains. The model can also be optimized with reinforcement learning.

**Ambiguous Citance**

In both cases, we use the BIOES labeling scheme for the output tags, following previous work which showed it outperforms other options (e.g., Ratinov and Roth, 2009).

**Abstract**

We analyze some of the fundamental design challenges and misconceptions that underlie the development of an efficient and robust NER system. In particular, we address issues such as the representation of text chunks, the inference approach needed to combine local NER decisions, the sources of prior knowledge and how to use them within an NER system. In the process of comparing several solutions to these challenges we reach some surprising conclusions, as well as develop an NER system that achieves 90.8 F1 score on the CoNLL-2003 NER shared task, the best reported result for this dataset.

**Vicuna** (Top 2 Relevant Paragraphs Summarized) (✔)

A simple technique of using word clusters generated from unlabeled text can improve performance of dependency parsing, Chinese word segmentation, and NER. The technique is based on word class models and uses a binary tree to represent words. The approach is related to distributional similarity, but not identical. The system's performance is significantly impacted by the choice of encoding scheme, and the less used BILOU formalism outperforms the widely adopted BIO tagging scheme.

Table 6: Examples of self-contained and ambiguous citances and human annotators' preference for summaries (✔). For the self-contained citance, the annotators found that the paper's abstract has a higher degree of coverage and better fit to the reading context. However, in the case of an ambiguous citance, the abstract was found insufficient to understand the citance because additional information had to be retrieved from the document. In this case, annotators preferred the generated contextualized summary over the abstract.

Our approach relies on LLMs that are constantly improved by the research community. The results of our experiments may vary with the introduction of newer LLMs. However, the underlying approach itself is intuitive and can be easily adapted to incorporate newer LLMs as they become available.

It is also important to recognize an important but often overlooked limitation of any summarization technology, namely the lack of a clear definition of what constitutes a good summary, given the its purpose. In our case, the purpose of the summary is to help readers understand the relevance of a citation without having to look up the cited paper. While we used abstracts as reference for comparison, our evaluation method does not induce true information needs relating to this purpose. This makes a fair comparison between abstracts and contextualized summaries difficult. In addition, the availability of expert annotators forced us to focus on the NLP field, which means that our results may not generalize to scientific papers from other fields. We hope that our work will encourage the research community to develop a more robust evaluation methodology for summarization tailored to the specific purposes of different types of summaries.

## Acknowledgments

This work was partially supported by the European Commission under grant agreement GA 101070014 (OpenWebSearch.eu). Computations for this work were done (in part) using resources of the Leipzig University Computing Center.

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

# A  Prompt Instructions

We used the following summarzation instructions to summarize the top-2 most relevant paragraphs:

1. Generate a coherent summary for the following scientific text in not more than 5 sentences.

2. Generate a short summary of the following scientific text. The summary should not be more than 5 sentences long.

3. Summarize the following scientific text in not more than 5 sentences.

By manually inspecting the model output of each instruction, we found that the first two instructions resulted in fluent summaries. However, excluding "coherent" from the instruction sometimes resulted in a bulleted list of important sentences in the summary. The third instruction aggressively compressed the content into a concise summary, but this resulted in a (partial) loss of information.

We used the following instructions to paraphrase the-5 most relevant sentences as a summary:

1. Generate a coherent paraphrased text for the following scientific text.

2. Generate a paraphrased text for the following scientific text.

3. Paraphrase the following scientific text.

4. Combine the following scientific text into a coherent and concise text.

A similar effect to the summarization prompts above was observed when the word "coherent" was excluded from the instructions. The last two prompts performed poorly, and in most cases simply returned the input text.

## B   Prompt Templates

- Generate a coherent summary for the following scientific text in not more than 5 sentences.

  scientific text: {input}

  summary:

- A chat between a curious user and an artificial intelligence assistant. The assistant knows how to summarize scientific text and the user will provide the scientific text for the assistant to summarize.

  USER: Generate a coherent summary for the following scientific text in not more than 5 sentences: "{input}"

  ASSISTANT:

- ### Instruction: A chat between a curious human and an artificial intelligence assistant. The assistant knows how to summarize scientific text and the user will provide the scientific text for the assistant to summarize.

  ### Input: Generate a coherent summary for the following scientific text in not more than 5 sentences: "{input}"

  ### Output:

For GPT4 we used the direct instruction prompt without template. The first template worked well only with the Alpaca and Falcon models, but not with the Vicuna and LLaMA-CoT models. If the last sentence of the input contained ':' or an abbreviation, these models tried to adapt the output to it. The Vicuna model presented summaries as an enumerated list. The LLaMA-CoT model did not "understand" this template and therefore did not produce any output. Therefore, we excluded this template from further experiments. Alpaca, LLaMA-CoT, and Falcon generated readable summaries based on the second template, but Vicuna did not, which prevented us from using it further. For example, the LLaMA-CoT model generated the following output when paraphrasing: "The text you provided is already paraphrased. It contains several sentences that express the same idea in different ways." The third prompt template yielded readable, coherent, and understandable summaries from all four models and caused artifacts in the summaries in only a few individual examples. Therefore, we used this prompt template with its associated instructions for our experiments.

## C   Summarization Models

1. Alpaca (Taori et al., 2023) was fine-tuned based on the LLaMA 7B model (Touvron et al., 2023) using 52,000 instruction-following examples (Wang et al., 2022).

2. Vicuna (Chiang et al., 2023) was fine-tuned based on LLaMA using user conversations from ShareGPT.[8] It has shown competitive effectiveness when evaluated with GPT4. We used the 13B variant.

3. LLaMA-CoT[9] was fine-tuned based on datasets inducing chains of thought (CoT) and logical reasoning (Qingyi Si, 2023).

4. Falcon (Almazrouei et al., 2023) was trained based on the RefinedWeb dataset (Penedo et al., 2023), which has been obtained by extensive filtering and deduplication of publicly available web data. At the time of writing, it is the state of the art on the open-llm leaderboard.[10] We used the 40B-Instruct variant.

---

[8] https://sharegpt.com/
[9] https://huggingface.co/ausboss/llama-30b-supercot
[10] https://huggingface.co/spaces/HuggingFaceH4/open_llm_leaderboard

5. GPT4 (Bubeck et al., 2023) is the latest version of OpenAI's popular GPT model class, showing the best effectiveness in several benchmarks. Therefore, we used it to bootstrap reference summaries for our summary evaluation (Section 5.2). Unlike the above open source models, it is accessible only through the OpenAI API.[11]

## D   Contextualized Summary Examples

Tables 7, 8, 9, and 10 show examples of generated contextualized summaries using alternative models.

---

[11] https://platform.openai.com/docs/models/GPT4

**Nematus: a Toolkit for Neural Machine Translation** (Sennrich et al., 2017)

```
Citance Context
```

Additionally, Nematus supports minimum risk training (MRT) (Shen et al., 2016) to optimize towards an arbitrary, sentence-level loss function.

Various MT metrics are supported as loss functions, including smoothed sentence-level BLEU (Chen and Cherry, 2014), METEOR (Denkowski and Lavie, 2011), BEER (Stanojevic and Sima'an, 2014), and any interpolation of implemented metrics.

To stabilize training, Nematus supports early stopping based on cross entropy or an arbitrary loss function defined by the user.

```
Abstract
```

BLEU is the de facto standard machine translation (MT) evaluation metric. However, because BLEU computes a geometric mean of n-gram precisions, it often correlates poorly with human judgment on the sentence-level. Therefore, several smoothing techniques have been proposed. This paper systematically compares 7 smoothing techniques for sentence-level BLEU. Three of them are first proposed in this paper, and they correlate better with human judgments on the sentence-level than other smoothing techniques. Moreover, we also compare the performance of using the 7 smoothing techniques in statistical machine translation tuning.

**Top 5 Relevant Sentences Paraphrased** (Query Context: *citance-SciBERT*)

```
GPT4
```

In this study, we conduct a systematic comparison of seven different smoothing methods for sentence-level BLEU. Although recent WMT Evaluation Task reports (Callison-Burch et al., 2011; Callison-Burch et al., 2012) have demonstrated that numerous other metrics have a stronger correlation with human evaluations of translation quality than BLEU, it continues to be the widely accepted standard for assessment and optimization. Smoothing 3 is incorporated into the official NIST evaluation toolkit (mteval-v13a.pl), and as a result, various smoothing approaches for sentence-level BLEU have been suggested (Lin and Och, 2004; Gao and He, 2013).

```
LLaMA-CoT
```

In this paper, we compare various smoothing techniques for sentence-level BLEU. Although other metrics are better at assessing translation quality, BLEU remains the standard evaluation metric. Smoothing 3 is implemented in the official NIST toolkit. Several smoothing techniques have been proposed.

**Top 2 Paragraphs Summarized** (Query Context:*similar-BM25*)

```
GPT4
```

Since its invention, BLEU has been the most widely used metric for machine translation evaluation and tuning. Despite other metrics correlating better with human judgments of translation quality, BLEU remains the de facto standard. This is likely because models trained using BLEU obtain the highest scores from humans and even from other metrics, making it the best tuning metric from a quality perspective.

```
Vicuna
```

BLEU is the most widely used metric for machine translation evaluation and tuning since its invention. Although other metrics correlate better with human judgments of translation quality, BLEU remains the de facto standard due to its effectiveness as a tuning metric, with models trained using BLEU obtaining the highest scores from both humans and other metrics.

Table 7: Examples of automatically generated contextualized summaries from the best models for both granularities (LLaMa and Vicuna). Summaries from GPT4 are used as ground truth for comparison after post processing for faithfulness and hallucinations.

**Nematus: a Toolkit for Neural Machine Translation** (Sennrich et al., 2017)

```
Citance Context
```

Additionally, Nematus supports minimum risk training (MRT) (Shen et al., 2016) to optimize towards an arbitrary, sentence-level loss function.

Various MT metrics are supported as loss functions, including smoothed sentence-level BLEU (Chen and Cherry, 2014), METEOR (Denkowski and Lavie, 2011), BEER (Stanojevic and Sima'an, 2014), and any interpolation of implemented metrics.

To stabilize training, Nematus supports early stopping based on cross entropy or an arbitrary loss function defined by the user.

```
Abstract
```

This paper describes Meteor 1.3, our submission to the 2011 EMNLP Workshop on Statistical Machine Translation automatic evaluation metric tasks. New metric features include improved text normalization, higher-precision paraphrase matching, and discrimination between content and function words. We include Ranking and Adequacy versions of the metric shown to have high correlation with human judgments of translation quality as well as a more balanced Tuning version shown to outperform BLEU in minimum error rate training for a phrase-based Urdu-English system.

**Top 5 Relevant Sentences Paraphrased** (Query Context: *citance-SciBERT*)

```
GPT4
```

In assessment tasks, the designed metrics typically aim to score the top translations from systems that are primarily optimized for BLEU. Meteor measures translation hypotheses by aligning them with reference translations and computing sentence-level similarity scores. The extra features in Meteor 1.3 enable more balanced parameters that allocate the responsibility of penalizing different kinds of incorrect translations. Since this task does not offer a devtest set, we choose a Meteor version by examining the efficiency of employing multiple metric versions to fine-tune phrase-based translation systems for the same language pair. We utilize Z-MERT to optimize this system for both baseline metrics and the Meteor versions mentioned in earlier sections.

```
LLaMA-CoT
```

In this study, we evaluate the performance of various metrics in translation evaluation tasks. We focus on the BLEU metric, which is commonly used to score 1-best translations from systems. We also explore the Meteor metric, which is designed to evaluate translation hypotheses by aligning them to reference translations and calculating sentence-level similarity scores. The new features in Meteor 1.3 allow for more balanced parameters, distributing responsibility for penalizing various types of erroneous translations. Since this task does not provide a devtest set, we select a version of the metric by exploring its effectiveness in tuning phrase-based translation systems for the same language pair. We use Z-MERT to tune this system to baseline metrics as well as the versions of Meteor discussed in previous sections.

**Top 2 Paragraphs Summarized** (Query Context:*similar-BM25*)

```
GPT4
```

A machine translation (MT) system tuning experiment was conducted on Urdu-English data to compare the effectiveness of using multiple versions of Meteor in minimum error rate training. Although versions tuned to various human judgments did not perform as well as the widely used BLEU metric, a balanced tuning version of Meteor consistently outperformed BLEU. The Meteor 1 metric has been shown to have a high correlation with human judgments in evaluations. However, previous versions of the metric were limited by lack of punctuation handling, noise in paraphrase matching, and lack of discrimination between word types. The introduction of new resources for all WMT languages, including text normalizers, filtered paraphrase tables, and function word lists, improved Meteor's correlation with human translation rankings and adequacy scores on unseen test data.

```
Vicuna
```

A team of researchers conducted an experiment to compare the effectiveness of using multiple versions of the Meteor machine translation system on Urdu-English data. They found that a balanced tuning version of Meteor consistently outperformed the widely used BLEU metric. The Meteor 1 metric, which has been shown to have high correlation with human judgments, was used in the experiment. However, previous versions of the metric had limitations such as lack of punctuation handling and discrimination between word types. The team introduced new resources for all WMT languages, including text normalizers, filtered paraphrase tables, and function word lists, which allowed them to tune versions of the metric that showed higher correlation with human translation rankings and adequacy scores on unseen test data. The evaluation resources were found to be modular and usable with any other evaluation metric or MT software.

Table 8: Examples of automatically generated contextualized summaries from the best models for both granularities (LLaMa and Vicuna). Summaries from GPT4 are used as ground truth for comparison after post processing for faithfulness and hallucinations.

**Incorporating Glosses into Neural Word Sense Disambiguation** (Luo et al., 2018)

**Citance Context**

Additionally, Nematus supports minimum risk training (MRT) (Shen et al., 2016) to optimize towards an arbitrary, sentence-level loss function.

In this paper, we propose a novel model GAS: a gloss-augmented WSD neural network which is a variant of the memory network (Sukhbaatar et al., 2015b; Kumar et al., 2016; Xiong et al., 2016).

In order to measure the inner relationship between glosses and context more accurately, we employ multiple passes operation within the memory as the re-reading process and adopt two memory updating mechanisms.', '• In order to model semantic relationship of context and glosses, we propose a glossaugmented neural network (GAS) in an improved memory network paradigm.

**Abstract**

Most tasks in natural language processing can be cast into question answering (QA) problems over language input. We introduce the dynamic memory network (DMN), a neural network architecture which processes input sequences and questions, forms episodic memories, and generates relevant answers. Questions trigger an iterative attention process which allows the model to condition its attention on the inputs and the result of previous iterations. These results are then reasoned over in a hierarchical recurrent sequence model to generate answers. The DMN can be trained end-to-end and obtains state-of-the-art results on several types of tasks and datasets: question answering (Facebook's bAbI dataset), text classification for sentiment analysis (Stanford Sentiment Treebank) and sequence modeling for part-of-speech tagging (WSJ-PTB). The training for these different tasks relies exclusively on trained word vector representations and input-question-answer triplets.

**Top 5 Relevant Sentences Paraphrased** (Query Context: *citance-SciBERT*)

**GPT4**

We present the Dynamic Memory Network (DMN), a neural network framework designed for general question-answering tasks, which is trained using raw input-question-answer triplets. Recent work by Weston et al. on memory networks (Weston et al., 2015b) emphasizes incorporating a memory component for natural language question answering. This model employs two extremely large and deep LSTMs to encode a sentence in one language and subsequently decode it in another language. The primary innovation of the DMN architecture lies in its episodic memory module. Kim (Kim, 2014) documents the previous state-of-the-art result, which is based on a convolutional neural network utilizing multiple word vector representations.

**LLaMA-CoT**

We present the Dynamic Memory Network (DMN), a neural network-based framework for general question answering tasks that is trained using raw input-question-answer triplets. Recent work by Weston et al. (Weston et al., 2015b) focuses on adding a memory component for natural language question answering. This model uses two extremely large and deep LSTMs to encode a sentence in one language and then decode the sentence in another language. The main novelty of the DMN architecture is its episodic memory module. Kim (Kim, 2014) reports the previous state-of-the-art result based on a convolutional neural network that uses multiple word vector representations.

**Top 2 Paragraphs Summarized** (Query Context:*similar-BM25*)

**GPT4**

The Dynamic Memory Network (DMN) is a neural network-based framework designed for general question-answering tasks, trained using raw input-question-answer triplets. It can solve various tasks such as sequence tagging, classification problems, sequence-to-sequence tasks, and question-answering tasks requiring transitive reasoning. While deep learning models like recursive neural networks have been applied to many NLP tasks, they lack the memory and question modules needed for more varied tasks and transitive reasoning. Recurrent neural networks have been successfully used in language modeling, speech recognition, and sentence generation from images. The sequence-to-sequence model used for machine translation is a special case of the DMN without a question and episodic memory, mapping an input sequence directly to an answer sequence.

**Vicuna**

We propose the Dynamic Memory Network (DMN), a neural network based framework for general question answering tasks that is trained using raw input-question-answer triplets. It can solve sequence tagging tasks, classification problems, sequence-to-sequence tasks, and question answering tasks that require transitive reasoning. The DMN has a memory module and a question module, which are absent in other deep learning models such as recursive neural networks and chain-structured recurrent neural networks. The DMN is based on a sequence-to-sequence model used for machine translation, but it has a memory and a question module.

Table 9: Examples of automatically generated contextualized summaries from the best models for both granularities (LLaMa and Vicuna). Summaries from GPT4 are used as ground truth for comparison after post processing for faithfulness and hallucinations.

**Efficient and Robust Question Answering from Minimal Context over Documents** (Min et al., 2018)

**Citance Context**

In addition, they cover different domains, including Wikipedia (Rajpurkar et al., 2016; Joshi et al., 2017), news (Hermann et al., 2015; Trischler et al., 2016), fictional stories (Richardson et al., 2013; Kočiský et al., 2017), and textbooks (Lai et al., 2017; Xie et al., 2017) .', 'Recently, researchers have developed largescale QA datasets, which requires answering the question over a large set of documents in a closed (Joshi et al., 2017) or open-domain (Dunn et al., 2017; Berant et al., 2013; Chen et al., 2017; Dhingra et al., 2017).

Following previous work on distant supervision (DS) for relation extraction (Mintz et al., 2009 ), we use a procedure to automatically associate paragraphs to such training examples, and then add these examples to our training set.

We use the following process for each questionanswer pair to build our training set.', 'If there is no paragraph with non-zero overlap, the example is discarded; otherwise we add each found pair to our DS training dataset.

**Abstract**

Modern models of relation extraction for tasks like ACE are based on supervised learning of relations from small hand-labeled corpora. We investigate an alternative paradigm that does not require labeled corpora, avoiding the domain dependence of ACEstyle algorithms, and allowing the use of corpora of any size. Our experiments use Freebase, a large semantic database of several thousand relations, to provide distant supervision. For each pair of entities that appears in some Freebase relation, we find all sentences containing those entities in a large unlabeled corpus and extract textual features to train a relation classifier. Our algorithm combines the advantages of supervised IE (combining 400,000 noisy pattern features in a probabilistic classifier) and unsupervised IE (extracting large numbers of relations from large corpora of any domain). Our model is able to extract 10,000 instances of 102 relations at a precision of 67.6%. We also analyze feature performance, showing that syntactic parse features are particularly helpful for relations that are ambiguous or lexically distant in their expression.

**Top 5 Relevant Sentences Paraphrased** (Query Context: *citance-SciBERT*)

**GPT4**

Our method employs Freebase (Bollacker et al., 2008), a vast semantic database, to offer distant supervision for extracting relationships. Since a single sentence might provide an inaccurate signal, our algorithm trains a multiclass logistic regression classifier, determining weights for each noisy characteristic. The core idea of our distant supervision technique is to utilize Freebase to supply a training set of relationships and entity pairs involved in those relationships. Each entity pair in every sentence of the test corpus undergoes feature extraction, and the regression classifier predicts a relationship name for each pair based on the features from all sentences in which it was present. For each entity pair, we consolidate the features from the numerous sentences in which the pair appeared into a single feature vector, enabling us to furnish our classifier with more data, leading to more precise labels.

**LLaMA-CoT**

Our algorithm uses Freebase, a large semantic database, to provide distant supervision for relation extraction. It trains a multiclass logistic regression classifier, learning weights for each noisy feature. The intuition of our distant supervision approach is to use Freebase to give us a training set of relations and entity pairs that participate in those relations. Each entity pair in the test corpus is run through feature extraction, and the regression classifier predicts a relation name for each entity pair based on the features from all of the sentences in which it appeared. For each pair of entities, we aggregate the features from the many different sentences in which that pair appeared into a single feature vector, allowing us to provide our classifier with more information, resulting in more accurate labels.

**Top 2 Paragraphs Summarized** (Query Context:*similar-BM25*)

**GPT4**

Distant supervision is an approach that uses Freebase, a large semantic database, to provide a training set of relations and entity pairs for relation extraction. The method identifies entities in sentences using a named entity tagger and extracts features from sentences containing instances of Freebase relations. This approach allows for the use of much larger amounts of data compared to supervised training paradigms, as it can utilize millions of Wikipedia articles and instances of relations connecting numerous entities. Combining vast numbers of features in a large classifier helps overcome problems with bad features. Distant supervision is an extension of the paradigm used for exploiting WordNet to extract hypernym relations between entities and is similar to the use of weakly labeled data in bioinformatics.

**Vicuna**

Our distant supervision approach uses Freebase to provide a training set of relations and entity pairs that participate in those relations. We use a named entity tagger to identify entities in sentences and extract features from sentences that contain two entities participating in a Freebase relation. Our algorithm is an extension of the paradigm used by Snow et al. (2005) and is similar to the use of weakly labeled data in bioinformatics. We use a logistic regression classifier to combine the vast numbers of features extracted from the text. Our approach can use much larger amounts of data than the supervised training paradigm, which uses only 17,000 relation instances as training data. We used 1.2 million Wikipedia articles and 1.8 million instances of 102 relations connecting 940,000 entities in our experiments. This helps to overcome the problem of bad features.

Table 10: Examples of automatically generated contextualized summaries from the best models for both granularities (LLaMa and Vicuna). Summaries from GPT4 are used as ground truth for comparison after post processing for faithfulness and hallucinations.