# OpenReview forum: "Citance-Contextualized Summarization of Scientific Papers"
_EMNLP/2023/Conference — EMNLP 2023 Findings_

### Official Review · Reviewer_mxe2 · 2023-08-03

**Soundness:** 3

**Excitement:**

3: Ambivalent: It has merits (e.g., it reports state-of-the-art results, the idea is nice), but there are key weaknesses (e.g., it describes incremental work), and it can significantly benefit from another round of revision. However, I won't object to accepting it if my co-reviewers champion it.

**Missing References:**

Faceted Summarization seems to be relevant (while not citance based) to the present work:

Meng, Rui, et al. "Bringing structure into summaries: a faceted summarization dataset for long scientific documents." arXiv preprint arXiv:2106.00130 (2021).

La Quatra, Moreno, Luca Cagliero, and Elena Baralis. "Exploiting pivot words to classify and summarize discourse facets of scientific papers." Scientometrics 125 (2020): 3139-3157.

Li, Lei, et al. "CIST@ CLSciSumm-19: Automatic Scientific Paper Summarization with Citances and Facets." BIRNDL@ SIGIR 54 (2019).

**Paper Topic And Main Contributions:**

This paper introduces a novel approach for contextualized summarization of scientific papers, focusing on generating informative summaries that depend on the citances (citation texts) in the citing paper. The main contribution is a method that grounds the summaries in the context of the citances, helping readers quickly find relevant information.

The paper also presents a large dataset from the computer science domain. The proposed approach utilizes LLMs to tailor the summaries to the citances, offering a potential solution for readers seeking specific connections between citing and cited papers.

The contributions of the paper are:
1. Introduction of a novel approach for contextualized summarization of scientific papers.
2. Generation of informative summaries dependent on the citances (citation texts).
3. Generation of a large dataset from the computer science domain.
4. Exploration of the use of Large Language Models (LLMs) to tailor summaries to the citances.

**Questions For The Authors:**

Question A: Regarding the dataset could you clarify if any summary is provided for the citance-cited_paper pair (summary length is 117 on avg for the provided dataset)? What metadata are included in the dataset?

Question B: In your evaluation, you focus solely on the use of LLMs. Have you considered including standard summarization models or extractive baselines in your comparison to provide a more comprehensive evaluation?

Question C: The abstract of a paper is found to be the most appreciated summary in the evaluation. How do you interpret this result in the context of your proposed citance-based summarization approach?

Question D: Could you elaborate on the choice of using GPT-4 generated ground truth summaries and manually verifying them? How do you ensure the reliability and lack of bias in the evaluation process, given that manual generation of ground truth summaries might offer a more robust evaluation?

Question E: The number of real evaluation data points for the summarization task seems limited, as only 15 papers and 25 randomly chosen citances are used. How representative do you believe this evaluation is for drawing broader conclusions about the effectiveness of the proposed approach? Have you considered expanding the evaluation dataset to ensure more robust results?

**Reasons To Accept:**

1. The paper introduces a new dataset. This dataset can be highly valuable for the NLP community, providing a new resource for research and development in the field of summarization and related areas.

2. The paper addresses an interesting and important task in the field of summarization - contextualized summarization of scientific papers. By focusing on generating summaries that are dependent on the citances in the citing paper, the proposed approach offers a novel perspective on summarization.

3. The evaluation methodology of the proposed approach is robust and considers different settings (within the LLMs scope).

**Reasons To Reject:**

1. One of the weaknesses of this paper is that the dataset provided, Context-SciSumm, lacks large-scale human-generated summaries. While the dataset contains a large number of citances and scientific papers, the absence of human-generated summaries can limit its usefulness for training deep learning models and evaluating against standard summarization benchmarks.
[EDIT: I better understand the difficulty of having annotations at scale.]

2. The authors' exploration in the paper is focused solely on the use of Large Language Models (LLMs) for contextualized summarization. The lack of comparison with standard summarization models or even extractive baselines raises concerns about the thoroughness of the evaluation and the performance of the proposed approach compared to existing methods.
[EDIT: the results reported in the response about other extractive/abstractive methods should be at least in the Appendix].

3. The paper's claim of providing informative summaries tailored to citances might be weakened by the fact that, in the end, the abstract is found to be the most appreciated summary. This discrepancy could raise questions about the effectiveness and utility of the proposed approach in practice.

4. The use of GPT-4 generated ground truth summaries, albeit manually verified, might raise doubts about the reliability and bias in the evaluation process. Manual generation of ground truth summaries could potentially lead to more accurate and unbiased evaluation results.

5. The paper lacks clarity regarding the number of real evaluation data points for the summarization task. The selection of only 15 papers from the ACL anthology and 25 randomly chosen citances for evaluation might not be representative enough to draw general conclusions about the effectiveness of the proposed approach.
[EDIT: I acknowledge the difficulty of having a strong set of annotators, but the number of papers seems to limited. Maybe the suggestion is to increase the number of papers while reducing the number of citances?]

**Reproducibility:**

3: Could reproduce the results with some difficulty. The settings of parameters are underspecified or subjectively determined; the training/evaluation data are not widely available.

**Reviewer Confidence:**

4: Quite sure. I tried to check the important points carefully. It's unlikely, though conceivable, that I missed something that should affect my ratings.

---

> ### Author Rebuttal · Authors · 2023-08-28
>
> Thanks for your detailed comments and questions. We first address the rejection reasons (enumerated as R1, R2 etc., ) followed by answers to specific questions.
>
> ### R1. Absence of human-generated summaries
>
> We acknowledge the desirability of having human written, high-quality summaries for each example in the corpus. However, just like in our case, achieving this ideal isn’t feasible for many third-party large scale summarization datasets either due to resource constraints and the inherent cognitive demands associated with the annotation. Notably, well-known datasets such as CNN DailyMail, XSum, and Arxiv adopt heuristics from the news web pages or already available abstracts instead of (new) human-written summaries for each example to train summarization models.
>
> Given our corpus's substantial scale with nearly 540K documents and 4.6 million citances, along with the manifold context variants for query-based summaries, manual annotation for all instances becomes prohibitively costly. Thus, as a foundational step, we validated several state-of-the-art large language models (LLMs) for zero-shot summarization, supported by rigorous automatic and manual evaluation of summary quality. Additionally, we introduce a benchmark comprising 100 summaries—typically standard for manual evaluations in summarization studies—assessed for coverage, coherence, and focus.
>
> The optimal model and query configurations we provide can now be harnessed by the community for dataset expansion, including the creation of domain-specific subsets (e.g., medicine, physics, computer science).
>
>
> ### R2. Sole focus on LLMs
> A main reason to investigate modern LLMs is their superior capability at generating high quality summaries in a zero-shot manner in comparison to trained supervised models (see table of automatic metric scores provided below). Notably, Goyal et al. (2022) News Summarization and Evaluation in the Era of GPT-3,  demonstrated the marked preference for GPT-3 summaries among humans, especially those generated using simple prompts. Unlike supervised models, LLMs are not prone to dataset-specific challenges, such as learning simplistic patterns or positional biases—a recurrent issue. Additionally, LLMs offer enhanced control over summary characteristics like length, style, and tone by fine-tuning their prompts.
>
> We consciously opted against extractive summarization due to its tendency to produce incoherent or excessively verbose outputs—sometimes even retaining numeric details or formulas verbatim from the target paper—which could undermine the intended seamless reading experience. In contrast, abstractive summaries address these concerns and contribute to a more cohesive output.
>
>
> ### R3. Abstract as a strong baseline
> Our paper aims to explore a new and relevant task—contextualized summarization, as the review noted. Our initial steps involved constructing a large-scale dataset, investigating SOTA LLMs, and comparing them to the familiar abstract baseline. Although annotators favored abstracts, it's likely because our dataset had more "general" citations not needing specific details from the target paper. This finding is important: it guides future choices on when to show abstracts (for general cases) or our summaries (for specific cases).
>
> We aim to further evaluate with more "specific" citations that demand details from the target paper. This will give us a clearer view of how well our contextualized summaries perform compared to abstracts.
>
> ### R4. Manual generation of ground truth summaries could potentially lead to more accurate and unbiased evaluation results.
> As mentioned in response to R2, prior work established that zero-shot summaries from GPT-3 were already better than the ground-truth summaries in multiple news datasets. The process of manually annotating summaries for papers in our case requires: (1) reading a citance, (2) the entire target paper, (3) fetching relevant content, (4) summarizing it within a fixed length budget. Moreover, this would still be an “incomplete set of summaries” as just one human reference might be insufficient (or even erroneous) for a given example. Clearly, this is infeasible from a budget perspective to do at scale. LLMs can offer multiple viable references in a cost-effective manner, even though human summaries are considered unbiased.
>
> Contrary to assuming no bias in human summaries, summarization heavily relies on background knowledge (education), topic familiarity, and the intended audience for which they are summarizing. For differences between expert and novice summarizers, see (1) Ruth Garner. Efficient text summarization costs and benefits. (2) Ann L Brown, Jeanne D Day, and Roberta S Jones. The development of plans for summarizing texts.
>
> Thus to balance for accuracy as well as feasibility, we bootstrapped the ground-truth generation by leveraging the SOTA GPT-4 model, followed by manual checks to identify potential errors.
>
> ### R5. Small sample size for manual evaluation
> Our evaluation encompassed 100 summaries, with each document receiving 3 judgments, yielding a total of 300 data points for result computation (lines 466 to 469).This is double in size compared to the evaluation by the related work CiteSum: Citation Text-guided Scientific Extreme Summarization and Domain Adaptation with Limited Supervision which evaluated only 50 examples, each with 2 annotators. While additional examples certainly hold value, our selection of 100 aligns with established norms in the field (see SummEval: Re-evaluating Summarization Evaluation).
>
> We now answer the direct questions.
>
> __Question A: Regarding the dataset could you clarify if any summary is provided for the citance-cited_paper pair (summary length is 117 on avg for the provided dataset)? What metadata are included in the dataset?__
>
> For each paper, the dataset contains (1) the citance text with its two contexts, (2) the extracted relevant content from the cited paper, and (3) three contextualized summaries. However, the summaries are only created for the 100 examples we manually evaluated to determine the best combination of citance context and LLM. The average summary length reported is over those bootstrapped by GPT4 for these 100 examples. Summaries for the remaining examples in the dataset will be provided as part of the final corpus release as the model inference for almost 13 million examples (4.6 million citances -> 3 citance contexts each, including citance as query -> 3 contextualized summaries including citance as query) is time and cost intensive for the computing resources at our disposal.
>
>
> The following JSON illustrates one complete example from our dataset.
>
> ```
> {
>   "citance_No": 5,
>   "citing_paper_id": 51871042,
>   "title": "DeepPavlov: Open-Source Library for Dialogue Systems",
>   "citing_paper_authors": "D Jason, Kavosh Williams, Geoffrey Asadi, Zweig",
>   "citance_marker": "(Williams et al., 2017)",
>   "citance_section": "Implemented Models and Skills",
>   "citance": "The skill implements Hybrid Code Networks (HCNs) described in (Williams et al., 2017).",
>   "prev_sentence": [
>     "Some of them are available for interactive online testing."
>   ],
>   "next_sentence": [
>     "The model is configurable: embeddings, slot filling component and intent classifier can be switched on and off on demand."
>   ],
>   "reference_paper_title": "Hybrid code networks: practical and efficient end-to-end dialog control with supervised and reinforcement learning",
>   "reference_paper_link": "13214003",
>   "top_2_paragraph_retrieval": {
>     "similar-BM25": "This paper presents a model for end-to-end learning, called Hybrid Code Networks (HCNs) which addresses these problems...",
>     "similar-keywords-BM25": "We compare to four past end-to-end approaches (Bordes and Weston, 2016; Liu and Perez, 2016; Eric and Manning, 2017; Seo et al., 2016) . We emphasize that past approaches have applied purely sequence-to-sequence models, or (as a baseline) purely programmed rules (Bordes and Weston, 2016)...",
>     "citance-SciBERT": "For optimization, we selected a policy gradient approach (Williams, 1992) , which has been successfully applied to dialog systems...",
>     "citance-BM25": "By contrast, Hybrid Code Networks are a hybrid of hand-coded rules and learned models. This paper presents a model for end-to-end learning, called Hybrid Code Networks (HCNs) which addresses these problems... "
>   },
>   "top_5_sentence_retrieval": {
>     "similar-BM25": "This paper presents a model for end-to-end learning, called Hybrid Code Networks (HCNs) which addresses these problems. By contrast, Hybrid Code Networks are a hybrid of hand-coded rules and learned models...",
>     "similar-keywords-BM25": "By contrast, Hybrid Code Networks are a hybrid of hand-coded rules and learned models. This paper presents a model for end-to-end learning, called Hybrid Code Networks (HCNs) which addresses these problems...",
>     "citance-SciBERT": "RL policy updates are made after each dia-log. Both the RNN and the developer code maintain state. HCNs also use an RNN to accumulate dialog state and choose actions. In each of these architectures, the RNN learns to manipulate entity values, for example by saving them in a memory...",
>     "citance-BM25": "By contrast, Hybrid Code Networks are a hybrid of hand-coded rules and learned models. This paper presents a model for end-to-end learning, called Hybrid Code Networks (HCNs) which addresses these problems..."
>   },
>   "contextualized_summaries": {
>     "abstract": "End-to-end learning of recurrent neural networks (RNNs) is an attractive solution for dialog systems; however, current techniques are data-intensive and require thousands of dialogs to learn simple behaviors. We introduce Hybrid Code Networks (HCNs), which combine an RNN with domain-specific knowledge encoded as software and system action templates...",
>     "GPT_citance_SciBert_top5": "After every dialogue, RL policy updates occur. State is maintained by both the RNN and the developer code. HCNs employ an RNN to gather dialogue state and select actions as well...",
>     "GPT_similar_BM25_top2": "This paper introduces Hybrid Code Networks (HCNs), a model for end-to-end learning that combines learning an RNN with domain knowledge expressed via software and action templates...",
>     "LLaMA_CoT_citance_SciBert_top5": "The RL policy is updated after each dialogue. Both the RNN and the developer code retain state. HCNs also use an RNN to accumulate dialogue state and select actions...",
>     "Vicuna_similar_BM25_top2": "This paper presents a new model called Hybrid Code Networks (HCNs) for end-to-end learning. HCNs allow developers to express domain knowledge and achieve better performance with less training data compared to existing techniques..."
>   }
> }
>
> ```
>
> __Question B: In your evaluation, you focus solely on the use of LLMs. Have you considered including standard summarization models or extractive baselines in your comparison to provide a more comprehensive evaluation?__
>
> We did experiment with multiple standard summarization models such as BART, T5, and BERT-Extractive. However, none of these models was close enough to the ground truth summaries as evaluated by automatic metrics (Table 3 in the paper). For completion, we provide the scores of these models for both retrieval contexts (top-2 paragraphs and top-5 sentences), alongside the LLMs.
>
>
> The tables below show the performance of standard summarization models (BART, T5, BERT-Extractive) compared to the LLMs. In most of the scenarios, standard summarization models are outperformed by the LLMs. Thus in our approach, we investigated LLMs extensively.
>
> Both tables are simple the Table 2 from the paper, extended by scores for the standard summarization models
>
> **Top-2 paragraphs retrieved as the to-be-summarized content from the cited paper.**
>
>
> | Model                 | BERTScore | R-1  | R-2  | R-L  |
> | ---------------------  | --------- | ---- | ---- | ---- |
> | ***similar-bm25***    |           |      |      |      |
> | __LLMs__              |           |      |      |      |
> | Alpaca                | 0.343     | 47.3 | 25.5 | 44.9 |
> | Falcon                | 0.401     | 48.2 | 27.1 | 45.0 |
> | LLaMA-CoT             | 0.448     | 53.0 | 31.9 | 50.5 |
> | Vicuna                | 0.465     | 58.7 | 35.4 | 55.8 |
> | __Standard Models__   |           |      |      |      |
> | BART                  | 0.249     | 39.9 | 18.7 | 37.5 |
> | Bert-Extractive       | 0.345     | 34.2 | 16.0 | 31.6 |
> | T5                    | 0.331     | 40.0 | 19.5 | 37.9 |
> | ***citance-SciBERT*** |           |      |      |      |
> | __LLMs__              |           |      |      |      |
> | Alpaca                | 0.390     | 54.3 | 32.2 | 52.0 |
> | Falcon                | 0.413     | 52.1 | 29.6 | 48.9 |
> | LLaMA-CoT             | 0.497     | 54.7 | 32.9 | 52.5 |
> | Vicuna                | 0.431     | 56.7 | 34.2 | 53.9 |
> | __Standard Models__   |           |      |      |      |
> | BART                  | 0.246     | 42.8 | 21.3 | 39.5 |
> | Bert-Extractive       | 0.334     | 35.8 | 19.6 | 33.7 |
> | T5                    | 0.378     | 46.8 | 23.4 | 44.2 |
>
> **Top-5 sentences retrieved as the to-be-summarized content from the cited paper.**
>
> | Model                 | BERTScore | R-1  | R-2  | R-L  |
> | --------------------- | --------- | ---- | ---- | ---- |
> | ***similar-bm25***    |           |      |      |      |
> | __LLMs__              |           |      |      |      |
> | Alpaca                | 0.616     | 56.2 | 35.4 | 54.8 |
> | Falcon                | 0.649     | 57.5 | 35.6 | 55.2 |
> | LLaMA-CoT             | 0.707     | 61.2 | 38.6 | 60.0 |
> | Vicuna                | 0.551     | 57.2 | 34.3 | 54.9 |
> | __Standard Models__   |           |      |      |      |
> | BART                  | 0.507     | 54.2 | 28.8 | 52.2 |
> | Bert-Extractive       | 0.571     | 38.7 | 21.0 | 37.4 |
> | T5                    | 0.344     | 36.8 | 16.5 | 35.5 |
> | ***citance-SciBERT*** |           |      |      |      |
> | __LLMs__              |           |      |      |      |
> | Alpaca                | 0.595     | 56.6 | 34.7 | 55.1 |
> | Falcon                | 0.656     | 56.8 | 36.2 | 55.3 |
> | LLaMA-CoT             | 0.748     | 62.9 | 40.6 | 60.9 |
> | Vicuna                | 0.607     | 58.8 | 36.0 | 56.6 |
> | __Standard Models__   |           |      |      |      |
> | BART                  | 0.534     | 57.3 | 31.4 | 55.2 |
> | Bert-Extractive       | 0.551     | 39.4 | 21.8 | 38.1 |
> | T5                    | 0.368     | 38.6 | 15.6 | 36.5 |
>
>
> __Question C: The abstract of a paper is found to be the most appreciated summary in the evaluation. How do you interpret this result in the context of your proposed citance-based summarization approach?__
>
> As described in the response to R3 of this review, we found the abstract to be a strong baseline in cases where the citance was generic and did not require an extensive reading of the cited paper from the reader. On the other hand, our contextualized summaries gained preference from annotators for specific citances.
>
> We acknowledge that this could have been better quantified in the paper and will do so. Also, we will perform a fine grained analysis of the citances and sample accordingly to balance generic and specific citances for which the contextualized summaries can be reevaluated together with the abstract.
>
> __Question D: Could you elaborate on the choice of using GPT-4 generated ground truth summaries and manually verifying them? How do you ensure the reliability and lack of bias in the evaluation process, given that manual generation of ground truth summaries might offer a more robust evaluation?__
>
> Please see the response to R4 above.
>
> __Question E: The number of real evaluation data points for the summarization task seems limited, as only 15 papers and 25 randomly chosen citances are used. How representative do you believe this evaluation is for drawing broader conclusions about the effectiveness of the proposed approach? Have you considered expanding the evaluation dataset to ensure more robust results?__
>
> In addition to the response provided in R5, it's important to elaborate that our choice to evaluate summaries exclusively from the ACL anthology stems from the availability of expert annotators within this specific domain. We recognize the necessity of a large and diverse annotator pool encompassing multiple domains for a robust evaluation across various subject areas. This is a direction we plan to pursue in the future.
>
> However, it's worth noting that our approach doesn't rely on domain-specific features for summarization. As such, it can be readily adapted to other domains without any customization.
>
>
> We hope that we have answered your questions in sufficient detail.

---

### Official Review · Reviewer_Gvqo · 2023-08-05

**Soundness:** 3

**Ethical Concerns:**

Yes

**Excitement:**

3: Ambivalent: It has merits (e.g., it reports state-of-the-art results, the idea is nice), but there are key weaknesses (e.g., it describes incremental work), and it can significantly benefit from another round of revision. However, I won't object to accepting it if my co-reviewers champion it.

**Justification For Ethical Concerns:**

Minor questions about the human volunteers

**Missing References:**

Not necessarily important, but perhaps as part of the discussion, Ball, P. Paper trail reveals references go unread by citing authors. Nature 420, 594 (2002). https://doi.org/10.1038/420594a is one of many papers that discusses the fact that authors may not fully read the papers they are citing - this tool might help alleviate (or detect!) that?

**Paper Topic And Main Contributions:**

The primary contribution of this paper is CONTEXT-SCISUMM a corpus of 4.6 Million citation texts over half a million academic papers in the CS domain. the dataset is introduced alongside the task of citance-focused abstract generation from papers cited in a work. The solution tot he latter task described in the paper compares a variety of retrieval methods and summarisation approaches, and evaluates the results.

**Questions For The Authors:**

1. was there any impact to multiple citations of the form "three systems do this [1,2,3]"?
2. more generally, are there properties of the surface form of the source and target that make this approach more effective for the user?
3. were the human annotators remunerated, and were they given opportunity to consent etc?

**Reasons To Accept:**

The ability to create task-tailored abstractive summarisations of this kind has applications beyond academic analysis. The dataset seems large and well-curated, and the example solution explores the possible solution space without fully answering it.

**Reasons To Reject:**

I think there is more work to be done on the comparative judgment - while initial work comparing this kind of summary to others is useful, a more targeted extrinsic evaluation might yield better performance in future work.

**Reproducibility:**

4: Could mostly reproduce the results, but there may be some variation because of sample variance or minor variations in their interpretation of the protocol or method.

**Reviewer Confidence:**

3: Pretty sure, but there's a chance I missed something. Although I have a good feel for this area in general, I did not carefully check the paper's details, e.g., the math, experimental design, or novelty.

---

> ### Author Rebuttal · Authors · 2023-08-28
>
> Thanks a lot for your comments and especially for the very interesting reference to the Ball, P. paper about unread citations. We answer your questions below.
>
> __was there any impact to multiple citations of the form "three systems do this [1,2,3]"?__
>
> The impact in such cases depends on how differently the three papers approach a given task. Moreover, our approach uses the citance text (and its various contexts) as the query to fetch relevant information, a citance that is too generic (e.g. “three systems do this…”) does not yield much informative queries. This subsequently results in the retrieval of unrelated or generic sentences that are summarized. In such cases, the abstract was more informative (as observed by our annotators).
>
> __more generally, are there properties of the surface form of the source and target that make this approach more effective for the user?__
>
> Continuing from the previous answer, our approach produces more informative and relevant summaries (compared to the abstract) where the citance is highly specific. So, we conjecture that a citance in the source paper must be rather specific to provide better search queries for finding relevant content from the target paper, consequently yielding superior summaries. Further investigation is warranted to consolidate this hypothesis.
>
> __were the human annotators remunerated, and were they given opportunity to consent etc?__
>
> The human annotators are full-time research assistants employed by our university's Computer Science faculty. They received proper compensation for their dedicated time towards research activities and have voluntarily given their consent to partake in the annotation process. We will clarify this in sufficient detail.

---

### Official Review · Reviewer_jGux · 2023-08-11

**Soundness:** 3

**Excitement:**

3: Ambivalent: It has merits (e.g., it reports state-of-the-art results, the idea is nice), but there are key weaknesses (e.g., it describes incremental work), and it can significantly benefit from another round of revision. However, I won't object to accepting it if my co-reviewers champion it.

**Paper Topic And Main Contributions:**

This paper is a summarization paper focusing on citances to enhance the exploration of relevant information from the cited paper.
In other words, this work aims to satisfy the readers who track citations and seek specific connections between the citing and cited papers.
they consider two additional contexts for a citance. first is the sentence including citance, and the sentence preceding and succeeding it. second, 2 most semantically similar sentences in the same paragraph.

**Questions For The Authors:**

How do you compare the proposed method with papers like (Cohan and Goharian, 2017)?
what is your result improvement and most highlighted novelty in comparison to the mentioned method? is this significant?

**Reasons To Accept:**

1. Proposing a large-scale dataset consisting of 540K documents and 4.6M citances.

**Reasons To Reject:**

1. Limited improvement,
2. Subjective idea, and intuition
3. limited ground truth has been generated by system, not human.

**Reproducibility:**

3: Could reproduce the results with some difficulty. The settings of parameters are underspecified or subjectively determined; the training/evaluation data are not widely available.

**Reviewer Confidence:**

3: Pretty sure, but there's a chance I missed something. Although I have a good feel for this area in general, I did not carefully check the paper's details, e.g., the math, experimental design, or novelty.

---

> ### Author Rebuttal · Authors · 2023-08-28
>
> Thanks for the comments. We try to answer your key questions below.
>
> __How do you compare the proposed method with papers like (Cohan and Goharian, 2017)?__
>
> Compared to Cohan and Goharian (2017), our method stands out in two distinct ways, as demonstrated in Figure 1 and elaborated from lines 209 to 221. Unlike their approach, which relies solely on verbatim citances for content retrieval, our investigation spans two additional citance contexts (lines 266 to 270), leading to 12 retrieval scenarios (listed in Table 2) that offer nuanced insights into context suitability for summarization. Moreover, while Cohan et al. consolidate all target paper content into a single summary for multiple citances, our approach crafts contextualized abstractive summaries independently for each citance, thus ensuring multiple, tailored summaries for the target paper.
>
> __What is your result improvement and most highlighted novelty in comparison to the mentioned method? is this significant?__
>
> Our approach's key innovation lies in generating multiple abstractive summaries for a cited paper, rooted in its various citances within a citing paper. To achieve this, we curated a sizable and fitting corpus, established clear baselines, and conducted comprehensive evaluations. Our study then substantiates the abstract's strength as a baseline through both automatic and qualitative assessments, spanning dimensions like "coverage," "focus," and "relevance."
>
> While our contextualized summaries didn't surpass the abstract in our results, this insight holds crucial implications for future investigations. It notably prompts an exploration of the interplay between citance specificity and preferred summary type. Our annotators highlighted that when citances were generic, requiring minimal reference to the cited paper, abstracts were preferred; however, for more specific citances, contextualized summaries proved superior.
>
> Previous work has consolidated all relevant information across citances to generate a single summary. Therefore, direct comparisons to such methods are neither possible nor meaningful.
>
> __“Limited ground truth has been generated by system, not human”__
>
> The system was used only to bootstrap the summarization process. The resulting summaries were manually reviewed and revised if necessary to ensure there are no hallucinations or factual errors (lines 472 to 475).

---

### Meta-Review · Area_Chair_q7ah · 2023-09-07

**Recommendation:** 3

**Metareview:**

This paper focuses on scientific paper summarization and investigates the grounding of generated summaries in the context of the citances (i.e. citation contexts), aiming at producing more informative summaries to assist readers in understanding a citation without the need to refer to the cited paper. Reviewers see many merits in the paper (adresses an interesting and timely task, new dataset, extensive experiments), and some of their initial concerns were addressed by the comprehensive authors's responses. After the discussion period, I feel that there are no real major issues with this work but reviewers have remaining concerns mainly about the evaluation methodology (e.g. GPT generated ground truth, limited scope of manual evaluation).

---

### Decision · Program_Chairs · 2023-10-07

**Decision:**

Accept-Findings

**Comment:**

This paper focuses on scientific paper summarization and investigates the grounding of generated summaries in the context of the citances (i.e. citation contexts), aiming at producing more informative summaries to assist readers in understanding a citation without the need to refer to the cited paper. Reviewers see many merits in the paper (adresses an interesting and timely task, new dataset, extensive experiments), and some of their initial concerns were addressed by the comprehensive authors's responses. After the discussion period, I feel that there are no real major issues with this work but reviewers have remaining concerns mainly about the evaluation methodology (e.g. GPT generated ground truth, limited scope of manual evaluation).